# Sequence generation with a physiologically plausible model of handwriting and Recurrent Mixture Density Networks

**Daniel Berio**[*1], **Memo Akten**[*1],

**Frederic Fol Leymarie**[1], **Mick Grierson**[1], and **Réjean Plamondon**[2]

## Abstract

The purpose of this study is to explore the feasibility and potential benefits of using a physiological plausible model of handwriting as a feature representation for sequence generation with recurrent mixture density networks. We build on recent results in handwriting prediction developed by Graves (2013), and we focus on generating sequences that possess the *statistical and dynamic qualities* of handwriting and calligraphic art forms. Rather than model raw sequence data, we first preprocess and reconstruct the input training data with a concise representation given by a motor plan (in the form of a coarse sequence of 'ballistic' targets) and corresponding dynamic parameters (which define the velocity and curvature of the pen-tip trajectory). This representation provides a number of advantages, such as enabling the system to learn from very few examples by introducing artificial variability in the training data, and mixing of visual and dynamic qualities learned from different datasets.

## 1 Introduction

Recent results (Graves, 2013) have demonstrated that, given a sufficiently large training data-set, *Long Short-Term Memory* (LSTM) (Hochreiter & Schmidhuber, 1997) Recurrent Mixture Density Networks (RMDNs) (Schuster, 1999) are capable of learning and generating convincing synthetic handwriting sequences. In this study we explore a similar network architecture combined with an intermediate feature representation, given by the parameters of a physiologically plausible model of handwriting: the Sigma Lognormal model (Plamondon, 1995; Plamondon et al., 2014).

In the work by Graves (2013) and subsequent derivations, the RMDN operates on raw sequences of points recorded with a digitizing device. In our approach we preprocess the training data using an intermediate representation that describes a form of "motor program" coupled with a sequence of dynamic parameters that describe the evolution of the pen tip. By doing so, we use a representation that is more concise (*i.e.* lower in dimensionality), meaningful (*i.e.* every data point is a high level segment descriptor of the trajectory), and is resolution independent.

This project stems from the observation that human handwriting results from the orchestration of a large number of motor and neural subsystems, and is ultimately produced with the execution of complex and skillful motions. As such we seek a representation that abstracts the complex task of trajectory formation from the neural network, which is then rather focused on a higher level task of movement planning. Note that for the scope of this study, we do not implement text-to-handwriting synthesis (Graves, 2013), but rather focus on the task of generating sequences that possess the statistical and dynamic qualities of handwriting, which can be expanded to calligraphy, asemic handwriting, drawings and graffiti (Berio & Leymarie, 2015; Berio et al., 2016)). In particular, we focus on two distinct tasks: (1) learning and generating motor plans and (2) given a motor plan,

---

[1]Goldsmiths College, University of London. Department of Computing.
[2]École Polytechnique de Montréal, Canada.

predicting the corresponding dynamic parameters that determine the visual and dynamic qualities of the pen trace. We then go on to show that this modular workflow can be exploited in ways such as: mixing of dynamic qualities between data-sets (a form of handwriting "style transfer" ) as well as learning from small datasets (a form of "one shot learning").

The remainder of this paper is organised as follows: in Section 2, after briefly summarising the background context, we then briefly describe the Sigma Lognormal model and RMDNs; in Section 3 we present the data preprocessing step and the RMDN models that build up our system; in Section 4 we propose various applications of the system, including learning handwriting representations from small datasets and mixing styles.

## 2 BACKGROUND

Our study is grounded on a number of notions and principles that have been observed in the general study of human movement as well as in the handwriting synthesis/analysis field (known as *Graphonomics* (Kao et al., 1986)). The speed profile of aiming movements is typically characterised by a "bell shape" that is variably skewed depending on the rapidity of the movement (Lestienne, 1979; Nagasaki, 1989; Plamondon et al., 2013). Complex movements can be described by the superimposition of a discrete number of "ballistic" units of motion, which in turn can each be represented by the classic bell shaped velocity profile and are often referred to as *strokes*. A number of methods synthesise handwriting through the temporal superimposition of strokes, the velocity profile of which is modelled with a variety of functions including sinusoidal functions (Morasso & Mussa Ivaldi, 1982; Maarse, 1987; Rosenbaum et al., 1995), Beta functions (Lee & Cho, 1998; Bezine et al., 2004), and lognormals (Plamondon et al., 2009). In this study we rely on a family of models known as the *Kinematic Theory of Rapid Human Movements*, that has been developed by Plamondon et al. in an extensive body of work since the 1990's (Plamondon, 1995; Plamondon et al., 2014). Plamondon et al. (2003) show that if we consider that a movement is the result of the parallel and hierarchical interaction of a large number of coupled linear systems, the impulse response of such a system to a centrally generated command asymptotically converges to a lognormal function. This assumption is attractive from a modelling perspective because it abstracts the high complexity of the neuromuscular system in charge of generating movements with a relatively simple mathematical model which further provides state of the art reconstruction of human velocity data (Rohrer & Hogan, 2006; Plamondon et al., 2013).

A number of methods have used neural inspired approaches for the task of handwriting trajectory formation (Schomaker, 1992; Bullock et al., 1993; Wada & Kawato, 1993). Similarly to our proposed method, Ltaief et al. (2012) train a neural on a preprocessed dataset where the raw input data is reconstructed in the form of handwriting model parameters. Nair & Hinton (2005) use a sequence of neural networks to learn the motion of two orthogonal mass spring systems from images of handwritten digits for classification purposes. With a similar motivation to ours, Plamondon & Privitera (1996) use a Self Organising Map (SOM) to learn a sequence of ballistic targets, which describe a coarse motor plan of handwriting trajectories. Our method builds in particular on the work of Graves (2013), who describes a system that uses a recurrent mixture density networks (RMDNs) (Bishop, 1994) extended with a LSTM architecture (Hochreiter & Schmidhuber, 1997), to generate synthetic handwriting in a variety of styles.

### 2.1 SIGMA LOGNORMAL MODEL

On the basis of Plamondon's Kinematic Theory (Plamondon, 1995), the Sigma Lognormal ($\Sigma\Lambda$) model (Plamondon & Djioua, 2006) describes complex handwriting trajectories via the vectorial superimposition of a discrete number of strokes. With the assumption that curved handwriting movements are done by rotating the wrist, the curvilinear evolution of strokes is described with a circular arc shape. Each stroke is charactersied by a variably assymmetric "bell shape" speed profile, which is described with a (3 parameter) lognormal function. The planar evolution of a trajectory is then described by a *sequence of virtual targets* $\{v_i\}_{i=1}^{i=m}$, which define "imaginary" (i.e. not necessarily located along the generated trajectory) loci at which each consecutive stroke is aimed. The virtual targets provide a low level description of the motor plan for the handwriting trajectory. A smooth trajectory is then generated by integrating the velocity of each stroke over time. The trajectory smoothness can be defined by adjusting the activation-time offset of a given stroke with respect to the

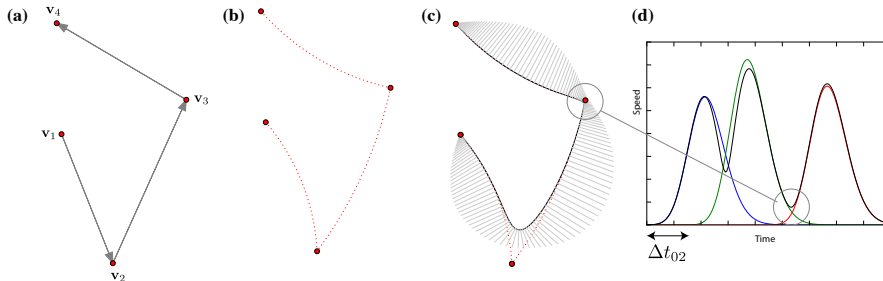

Figure 1: A sequence of virtual targets and the corresponding $\Sigma\Lambda$ trajectory. **(a)**, the virtual targets and the corresponding stroke aiming directions. **(b)**, the virtual targets and the corresponding circular arcs. **(c)**, a possible trajectory generated over the given sequence of virtual targets. While the generated trajectory might appear similar to a polynomial curve such as a B-Spline, it also describes a smooth and physiologically plausible velocity profile **(d)**.

previous stroke, which is denoted with $\Delta t_{0i}$; a smaller time offset (i.e. a greater overlap between lognormal components) will result in a smoother trajectory (Fig. 1, c). The curvature of the trajectory can be varied by adjusting the central angle of each circular arc, which is denoted with $\theta_i$. Equations and further details for the $\Sigma\Lambda$ model can be found in Appendix A.

A sequence of virtual targets provides a very sparse spatial description or "motor plan" for the trajectory evolution. The remaining stroke parameters, $\Delta t_{0i}$ and $\theta_i$, define the temporal, dynamic and geometric features of the trajectory and we refer to those as *dynamic parameters*.

## 2.2 RECURRENT MIXTURE DENSITY NETWORKS

*Mixture Density Networks (MDN)* were introduced by Bishop (1994) in order to model and predict the parameters of a *Gaussian Mixture Model (GMM)*, i.e. a set of means, covariances and mixture weights. Schuster (1999) showed that MDNs could be to model temporal data using RNNs. The author used *Recurrent Mixture Density Networks (RMDN)* to model the statistical properties of speech, and they were found to be more successful than traditional GMMs. Graves (2013) used LSTM RMDNs to model and synthesise online handwriting, providing the basis for extensions to the method, also used in Ha et al. (2016); Zhang et al. (2016). Note that in the case of a sequential model, the RMDN outputs a *unique set of GMM parameters for each timestep $t$*, allowing the probability distribution to change with time as the input sequence develops. Further details can be found in Appendix C.1.

## 3 METHOD

We operate on discrete and temporally ordered sequences of planar coordinates. Similarly to Graves (2013), most of our results come from experiments made on the IAM *online* handwriting database (Marti & Bunke, 2002). However, we have made preliminary experiments with other datasets, such as the Graffiti Analysis Database (Lab, 2009) as well as limited samples collected in our laboratory from a user with a digitiser tablet.

As a first step, we preprocess the raw data and reconstruct it in the form of $\Sigma\Lambda$ model parameters Section 3.1. We then train and evaluate a number of RMDN models for two distinct tasks:

1. **Virtual target prediction**. We use the *V2V-model* for this task. Given a sequence of virtual targets, this model predicts the next virtual target.

2. **Dynamic parameter prediction**. For this task we trained and compared two model architectures. Given a sequence of virtual targets, the task of these models is to predict the corresponding dynamic parameters. The *V2D-model* is condititioned only on the previous virtual targets, whereas the *A2D-model* is conditioned on both the previous virtual targets and dynamic parameters.

We then exploit the modularity of this system to conduct various experiments, details of which can found in Section 4.

### 3.1 PREPROCESSING: RECONSTRUCTING $\Sigma\Lambda$ PARAMETERS

A number of methods have been developed by Plamondon et. al in order to reconstruct $\Sigma\Lambda$-model parameters from digitised pen input data (O'Reilly & Plamondon, 2008; Plamondon et al., 2014; Fischer et al., 2014). These methods provide the ideal reconstruction of model parameters, given a high resolution digitised pen trace. While such methods are superior for handwriting analysis and biometric purposes, we opt for a less precise method (Berio & Leymarie, 2015) that is less sensitive to sampling quality and is aimed at generating virtual target sequences that remain perceptually similar to the original trace. We purposely choose to ignore the original dynamics of the input, and base the method on a geometric input data only. This is done in order to work with training sequences that are independent of sampling rate, and in sight of future developments in which we intend to extract handwriting traces from bitmaps, inferring causal/dynamic information from a static input as humans are capable of (Edelman & Flash, 1987; Freedberg & Gallese, 2007).

Our method operates on a uniformly sampled input contour, which is then segmented in correspondence with perceptually salient *key points*: loci of curvature extrema modulated by neighbouring contour segments (Brault & Plamondon, 1993; Berio & Leymarie, 2015), which gives an initial estimate of each virtual target $v_i$. We then (i) fit a circular arc to each contour segment in order to estimate the $\theta_i$ parameters and (ii) estimate the $\Delta t_{0i}$ parameters by analysing the contour curvature in the region of each key point. Finally, (iii) we iteratively adjust the virtual target positions to minimise the error between the original trajectory and the one generated by the corresponding $\Sigma\Lambda$ parameters. For Further details on the $\Sigma\Lambda$ parameter reconstruction method, the reader is referred to Appendix B.

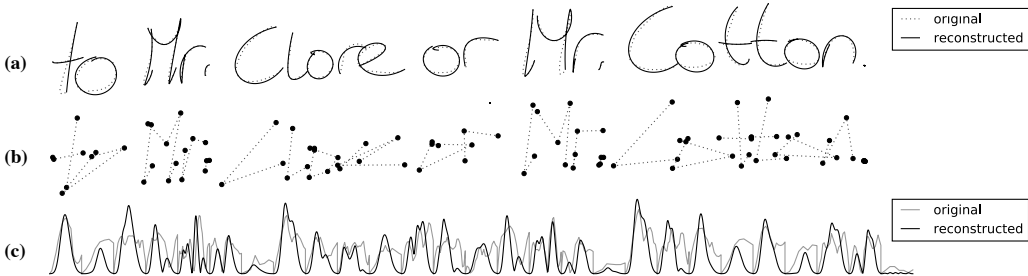

Figure 2: $\Sigma\Lambda$ parameter reconstruction. (a) The original and reconstructed trajectories. (b) The reconstructed virtual targets. Note that the virtual targets define a shape that is perceptually similar to the input. (c) Aligned and scaled speed profiles of the original (gray) and reconstructed (black) trajectories. Although the dynamic information in the input is ignored (due to uniform sampling), the two speed profiles show similarities in number and relative-height of peaks.

### 3.2 DATA AUGMENTATION

We can exploit the $\Sigma\Lambda$ parameterisation to generate many variations over a single trajectory, which are visually consistent with the original, and with a variability that is similar to the one that would be seen in multiple instances of handwriting made by the same writer (Fig. 3) (Djioua & Plamondon, 2008a; Fischer et al., 2014; Berio & Leymarie, 2015). Given a dataset of $n$ training samples, we randomly perturb the virtual target positions and dynamic parameters of each sample $n_p$ times, which results in a new augmented dataset of size $n + n \times n_p$ where legibility and trajectory smoothness is maintained across samples. This would not be possible on the raw online dataset, as perturbations for each data-point would eventually result in a noisy trajectory.

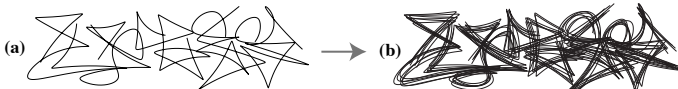

Figure 3: Data augmentation step.

### 3.3 PREDICTING VIRTUAL TARGETS WITH THE V2V-MODEL

The *V2V-model* is conditioned on a history of virtual targets and given a new virtual target it predicts the next virtual target (hence the name *V2V*). Note that each virtual target includes the corresponding

pen state — *up* (not touching the paper) or *down* (touching the paper). Repeatedly feeding the predicted virtual target back into the model at every timestep allows the model to synthesise sequences of arbitrary length. The implementation of this model is very similar to the handwriting prediction demonstrated by Graves (2013), although instead of operating directly on the digitised pen positions, we operate on the much coarser virtual target sequences which are extracted during the preprocessing step. The details of this model can be found in Appendix C.3

### 3.4 Predicting dynamic parameters with the V2D and A2D models

The goal of these models is to predict the corresponding dynamic parameters $(\Delta t_{0i}, \theta_i)$ for a given sequence of virtual targets. We train and compare two model architectures for this task. The *V2D-model* is conditioned on the history of virtual targets, and given a new virtual target, this model predicts the corresponding dynamic parameters $(\Delta t_{0i}, \theta_i)$ for the current stroke (hence the name *V2D*). Running this model incrementally for every stroke of a given virtual target sequence allows us to predict dynamic parameters for each stroke. The implementation of this model is very similar to the V2V-model, and details can be found in Appendix C.4.

At each timestep, the V2D model outputs and maintains internal memory of a probability distribution for the predicted dynamic parameters. However, the network has no knowledge of the parameters that are sampled and used. Hence, dynamic parameters might not be consistent across timesteps. This problem can be overcome by feeding the sampled dynamic parameters back into the model at the next timestep. From a human motor planning perspective this makes sense as, for a given drawing style, when we decide the curvature and smoothness of a stroke we will take into consideration the choices made in previously executed strokes.

The A2D model predicts the corresponding dynamic parameters $(\Delta t_{0i}, \theta_i)$ for the current stroke conditioned on a history of both virtual targets *and dynamic parameters* (i.e. all $\Sigma\Lambda$ parameters - hence the name *A2D*). We use this model in a similar way to the V2D model, whereby we run it incrementally for every stroke of a given virtual target sequence. However, internally, at every timestep the predicted dynamic parameters are fed back into the model at the next timestep along with the virtual target from the given sequence. The details of this implementation can be found in Appendix C.5.

## 4 Experiments and Results

**Predicting Virtual Targets.** In a first experiment we use the V2V model, trained on the preprocessed IAM dataset, to predict sequences of virtual targets. We prime the network by first feeding it a sequence from the test dataset. This conditions the network to predict sequences that are similar to the prime. We can see from the results (Fig. 4) that the network is indeed able to produce sequences that capture the statistical qualities of the priming sequence, such as overall incline, proportions, and oscillation frequency. On the other hand, we observe that amongst the generated sequences, there are often patterns which do not represent recognisable letters or words. This can be explained by the high variability of samples contained in the IAM dataset, and by the fact that our representation is very concise, with each data-point containing high significance. As a result, the slightest variation in a prediction is likely to cause a large error in the next. To overcome this problem, we train a new model with a dataset augmented with $10\times$ variations as described in Section 3.2. Due to our limited computing resources [1], we test this method on $1/10$th of the dataset, which results in a new dataset with the same size as the original, but with a lower number of handwriting specimens with a number of subtle variations per specimen. With this approach, the network predictions maintain statistical similarity with the priming sequences, and patterns emerge that are more evocative of letters of the alphabet or whole words, with fewer unrecognizable patterns (Fig. 4). To validate this result, we also test the model's performance training it on $1/10$th of the dataset, without data augmentation, and the results are clearly inferior to the previous two models. This suggests that the data augmentation step is highly beneficial to the performance of the network.

---

[1] We are thus not able to thoroughly test the large network architectures that would be necessary to train on the whole augmented dataset.

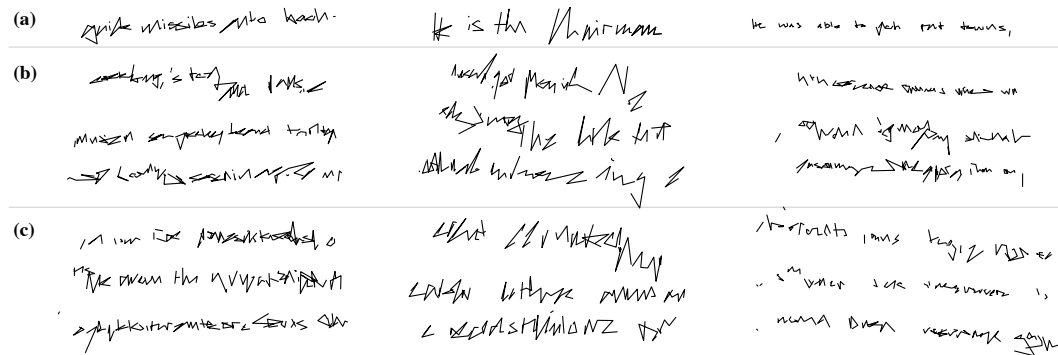

Figure 4: Predicting virtual targets. (a) Virtual targets from the test set (not seen during V2V training) used to prime the V2V models. (b) Sequences generated with the V2V model. (c) Sequences generated with the *augmented* V2V model. Note that the non-augmented V2V model produces more undesired 'errors'. This is more visibly noticable when rendered with dynamic parameters (Fig. 6).

**Predicting Dynamic Parameters.** We first evaluate the performance of both the V2D and A2D models on virtual targets extracted from the test set. Remarkably, although the networks have not been trained on these sequences, both models predict dynamic parameters that result in trajectories that are readable, and are often similar to the target sample. We settle on the A2D model trained on a $3\times$ augmented dataset, which we qualitatively assess to produce the best results (Fig. 5).

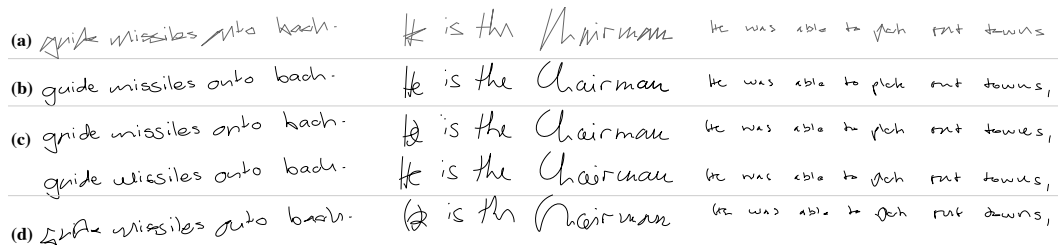

Figure 5: Dynamic parameter prediction. (a) Virtual targets from samples in the test set (not seen during training). (b) The original trajectories provided for comparison. (c) Trajectories reconstructed using predicted dynamic parameters. (d) Trajectories reconstructed with random dynamic parameters provided for comparison.

We then proceed with applying the same A2D model on virtual targets generated by the V2V models primed on the test set. We observe that the predictions on sequences generated with the augmented dataset are highly evocative of handwriting and are clearly different depending on the priming sequence (Fig. 6, c), while the predictions made with the non-augmented dataset are more likely to resemble random scribbles rather than human readable handwriting (Fig. 6, b). This further confirms the utility of the data augmentation step.

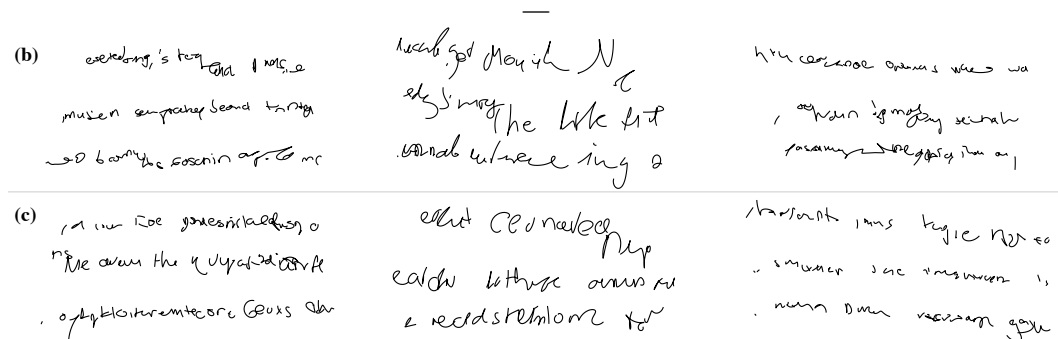

Figure 6: Trajectories reconstructed with dynamic parameters predicted for generated virtual targets from Fig. 4 using (b) non-augmented V2V, (c) augmented V2V.

**User defined virtual targets.** The dynamic parameter prediction models can also be used in combination with user defined virtual target sequences (Fig. 7). Such a method can be used to quickly and interactively generate handwriting trajectories in a given style, by a simple point and click procedure. The style (in terms of curvature and dynamics) of the generated trajectory is determined by the data used to train the A2D model, and by priming the A2D model with different samples, we can apply different styles to the user defined virtual targets.

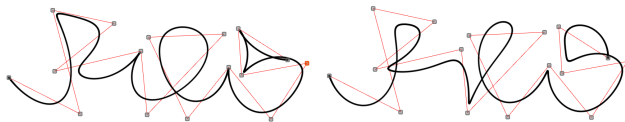

Figure 7: Dynamic parameters generated over user specified virtual targets for the word 'Res', using the A2D model trained on the IAM database.

**One shot learning.** In a subsequent experiment, we apply the data augmentaion method described in Section 3.2 to enable both virtual target and dynamic prediction models to learn from a small dataset of calligraphic samples recorded by a user using a digitiser tablet. We observe that with a low number of augmentations ($50\times$) the models generate quasi-random outputs, and seem to learn only the left to right trend of the input. With higher augmentation ($700\times$), the system generates outputs that are consistent to the human eye with the input data (Fig. 8). We also train our models using only a single sample (augmented $7000\times$) and again observe that the model is able to reproduce novel sequences that are similar to the input sample (Fig. 9). Naturally, the output is a form of recombination of the input, but this is sufficient to synthesise novel outputs that are qualitatively similar to the input. It should be noted that we are judging the performance of the one-shot learned models qualitatively, and we may not be testing the full limits of how well the models are able to generalise. On the other hand, these results, as well as the "style transfer" capabilities exposed in following section suggest a certain degree of generalisation.

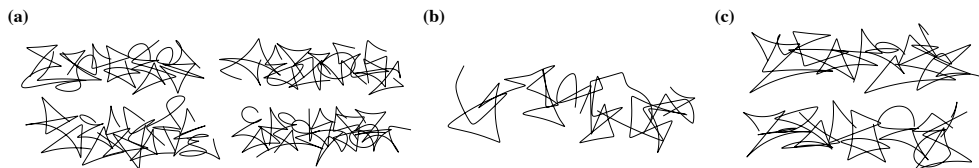

Figure 8: Training with small ($n = 4$) datasets. (a) Training set with $4$ samples. (b) Output of the networks when using $50\times$ data augmentation. (c) Output of the networks with $700\times$ data augmentation.

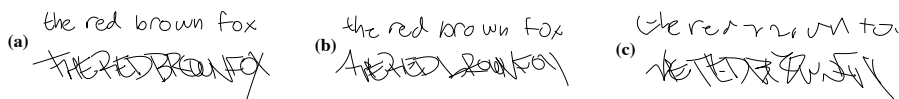

Figure 9: Training with single training samples. For each row: (a) Training sample (augmented $\times7000$). (b) Output of combined V2V/A2D models primed on the training sample. (c) Output without priming.

**Style Transfer.** Here, with a slight abuse of terminology, we utilise the term "style" to refer to the dynamic and geometric features (such as pen-tip acceleration and curvature) that determine the visual qualities of a handwriting trajectory. Given a sequence of virtual targets generated with the V2V model trained on one dataset, we can also predict the corresponding dynamic parameters with the A2D model trained on another. The result is an output that is similar to one dataset in lettering structure, but possesses the fine dynamic and geometric features of the other. If we visually inspect Fig. 10, we can see that both the sequence of virtual targets reconstructed by the dataset preprocessing method, and the trajectory generated over the same sequence of virtual targets with dynamic parameters learned from a different datasets, are both readable. This emphasises the importance of using perceptually salient points along the input for estimating key-points in the data-set preprocessing step (Section 3.1).

Furthermore, we can perform the same type of operation within a single dataset, by priming the A2D model with the dynamic parameters of a particular training example, while feeding it with the virtual targets of another. To test this we train both (V2V, A2D) models on a corpus containing 5 samples of the same sentence written in different styles and then augmented $1400\times$ (Fig. 11). We envision the utility of such as system in combination with virtual targets interactively specified by a user.

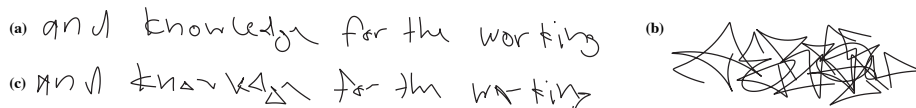

Figure 10: Style transfer mixing training sets. (a) The priming sequence from the V2V dataset (IAM). (b) A2D is trained on a different, single user specified sample. (c) The virtual targets from (a) rendered with the dynamic parameters predicted form the A2D model from (b).

Figure 11: Style transfer using priming. The leftmost column shows the entire training set consisting of 5 user drawn samples. The top row (slightly greyed out) shows the virtual targets for two of the training examples. Each cell in the table shows the corresponding virtual targets rendered using the dynamic parameters predicted with the A2D model primed with the sample in the corresponding row.

## 5 CONCLUSIONS AND FUTURE WORK

We have presented a system that is able to learn the parameters for a physiologically plausible model of handwriting from an online dataset. We hypothesise that such a movement centric approach is advantageous as a feature representation for a number of reasons. Using such a representation provides a performance that is similar to the handwriting prediction demonstrated by Graves (2013) and Ha et al. (2016), with a number of additional benefits. These include the ability to: (i) capture both the *geometry* and *dynamics* of a hand drawn/written trace with a single representation, (ii) express the variability of different types of movement concisely at the feature level, (iii) demonstrate greater flexibility for procedural manipulations of the output, (iv) mix "styles" (applying curvature and dynamic properties from one example, to the motor plan of another), (v) learn a generative model from a small number of samples ($n < 5$), (vi) generate resolution independent outputs.

The reported work provides a solid basis for a number of different future research avenues. As a first extension, we plan to implement the label/text input alignment method described in Graves' original work that should allow us to synthesise readable handwritten text and also to provide a more thorough comparison of the two methods. Our method strongly relies on an accurate reconstruction of the input in the preprocessing step. Improvements should target especially parts of the latter method that depend on user tuned parameters, such as the identification of salient points along the input (which requires a final peak detection pass), and measuring the sharpness of the input in correspondence with salient points.

ACKNOWLEDGMENTS

The system takes as a starting point the original work developed by (Graves, 2013). We use Tensorflow, the open-source software library for numerical computation and deep learning (Abadi et al., 2015), and a rapid implementation was possible thanks to a public domain implementation developed by (Ha, 2015).

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

## A    SIGMA LOGNORMAL MODEL

The Sigma Lognormal model (Plamondon & Djioua, 2006) describes complex handwriting trajectories via the vectorial superimposition of lognormal strokes. The corresponding speed profile $\Lambda_i(t)$ assumes a variably asymmetric "bell shape" which is described with a 3 parameter lognormal function

$$\Lambda_i(t) = -\frac{1}{\sigma_i\sqrt{2\pi}(t - t_{0i})}\exp\left(\frac{(ln(t - t_{0i}) - \mu_i)^2}{2\sigma_i{}^2}\right) \tag{1}$$

where $t_{0i}$ defines the activation time of a stroke and the parameters $\mu_i$ and $\sigma_i$ determine the shape of the lognormal function. $\mu_i$ is referred to as *log-time delay* and is biologically interpreted as the rapidity of the neuromuscular system to react to an impulse generated by the central nervous system (Plamondon et al., 2003); $\sigma_i$ is referred to as *log-response time* and determines the spread and asymmetry of the lognormal.

The curvilinear evolution of strokes is described with a circular arc shape, which results in

$$\phi_i(t) = \theta_i + \theta_i \left[ 1 + \mathrm{erf}\left( \frac{\ln(t - t_{0i}) - \mu_i}{\sigma_i \sqrt{2}} \right) \right],\tag{2}$$

where $\theta_i$ is the central angle of the circular arc that defines the shape of the $i^{\text{th}}$ stroke.

The planar evolution of a trajectory is defined by a *sequence of virtual targets* $\{\boldsymbol{v}_i\}_{i=1}^{i=m}$, where a trajectory with $m$ virtual targets will be characterised by $m - 1$ circular arc strokes. A $\Sigma\Lambda$ trajectory, parameterised by the virtual target positions, is given by

$$\boldsymbol{\xi}(t) = \boldsymbol{v}_1 + \int_0^t d\tau \Lambda_i(\tau) \sum_{i=1}^{m-1} \boldsymbol{\Phi}_i(\tau)\,(\boldsymbol{v}_{i+1} - \boldsymbol{v}_i),\tag{3}$$

$$\text{with}\quad \boldsymbol{\Phi}_i(t) = \begin{bmatrix} h(\theta_i)\cos\phi_i(t) & -h(\theta_i)\sin\phi_i(t) \\ h(\theta_i)\sin\phi_i(t) & -h(\theta_i)\cos\phi_i(t) \end{bmatrix},\quad \text{and}\quad h(\theta_i) = \begin{cases} \frac{2\theta_i}{2\sin\theta_i} & \text{if } |\sin\theta_i| > 0, \\ 1 & \text{otherwise,} \end{cases}\tag{4}$$

which scales the extent of the stroke based on the ratio between the perimeter and the chord length of the circular arc.

**Intermediate parameterisation.**    In order to facilitate the precise specification of timing and profile shape of each stroke, we recur to an intermediate parametrisation that takes advantage of a few known properties of the lognormal (Djioua & Plamondon, 2008b) in order to define each stroke with (i) a time offset $\Delta t_i$ with respect to the previous stroke, (ii) a stroke duration $T_i$ and (iii) a shape parameter $\alpha_i$, which defines the skewedness of the lognormal. The corresponding $\Sigma\Lambda$ parameters $\{t_{0i}, \mu_i, \sigma_i\}$ can be then computed with:

$$\sigma_i = \ln(1 + \alpha_i),\tag{5}$$

$$\mu_i = -\ln\left( -\frac{e^{-3\sigma_i} - e^{3\sigma_i}}{T_i} \right),\tag{6}$$

and

$$t_{0i} = t_{1i} - e^{\mu - 3\sigma} \qquad t_{1i} = t_{1(i-1)} + \Delta t_i \qquad t_{1(0)} = 0,\tag{7}$$

where $t_{1i}$ is the onset time of the lognormal stroke profile. As $\alpha$ approaches $0$, the shape of the lognormal converges to a Gaussian, with mean $t_1 + e^{\mu - \sigma^2}$ (the mode of the lognormal) and standard deviation $\frac{d}{6}$.

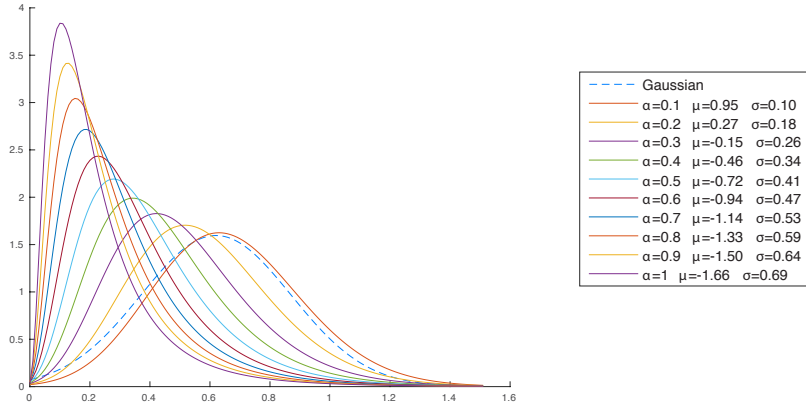

Figure 12: Lognormals with varying "skeweness" parameter $\alpha$ and corresponding values for $\mu, \sigma$. As $\alpha \to 0$, the lognormal approaches a Gaussian.

# B    RECONSTRUCTING $\Sigma\Lambda$ PARAMETERS FROM AN ONLINE DATASET

The $\Sigma\Lambda$ parameter reconstruction method operates on a input contour uniformly sampled at a fixed distance which is defined depending on the extent of the input, where we denote the $k$th sampled point along the input with $\boldsymbol{p}[k]$. The input contour is then segmented in correspondence with perceptually salient *key points*, which correspond with loci of curvature extrema modulated by neighbouring contour segments (Brault & Plamondon, 1993; Berio & Leymarie, 2015). The proposed approach shares strong similarities with previous work done for (i) compressing online handwriting data with a circular-arc based segmentation (Li et al., 1998) and (ii) for generating synthetic data for handwriting recognisers (Varga et al., 2005). The parameter reconstruction algorithm can be summarised with the following steps:

- Find $m$ key-points in the input contour.
- Fit a circular arc to each contour segment defined between two consecutive key-points (defining individual strokes), and obtain an estimate of each curvature parameter $\theta_i$.
- For each stroke compute the corresponding $\Delta t_i$ parameter by analysing the curvature signal in the region of the corresponding key-point.
- Define an initial sequence of virtual targets with $m$ positions corresponding with each input key-point.
- Repeat the following until convergence or until a maximum number of iterations is reached Berio & Leymarie (2015):
    - Integrate the $\Sigma\Lambda$ trajectory with the current parameter estimate.
    - Identify $m$ key-points in the generated trajectory.
    - Move the virtual target positions to minimise the distance between the key-points of the generated trajectory and the key-points on the input contour.

The details for each step are highlighted in the following paragraphs.

**Estimating input key-points.**    Finding significant curvature extrema (which can be counted as convex and concave features for a closed/solid shape) is an active area of research, as relying on discrete curvature measurements remains challenging. We currently rely on a method described by Feldman & Singh (2005), and supported experimentally by De Winter & Wagemans (2008): first we measure the turning angle at each position of the input $\boldsymbol{p}[k]$ and then compute a smooth version of the signal by convolving it with a Hanning window. We assume that the turning angles have

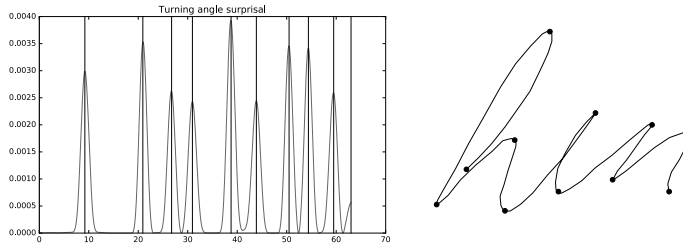

Figure 13: Input key-point estimation. Left, the (smoothed) turning angle surprisal signal and the key-points estimated with peak detection. Right, the corresponding key-points along the input trajectory.

been generated by a random process with a Von Mises distribution with mean at $0$ degrees, which corresponds with giving maximum probability to a straight line. We then measure the *surprisal* (i.e. the negative logarithm of the probability) for each sample as defined by Feldman & Singh (2005), which normalised to the $[0, 1]$ range simplifies to:

$$1 - \cos(\theta[k]), \tag{8}$$

where $\theta[k]$ is the (smoothed) turning angle. The first and last sample indices of the surprisal signal together with its local maxima results in $m$ key-point indices $\{\hat{z}_i\}$. The corresponding key-points along the input contour are then given by $\{\boldsymbol{p}\,[\hat{z}_i]\}$.

**Estimating stroke curvature parameters.** For each section of the input contour defined between two consecutive key-points, we estimate the corresponding stroke curvature parameter $\theta_i$ by first computing a least square fit of a circle to the contour section. We then compute the internal angle of the arc supported between the two key-points, which is equal $2\theta_i$, *i.e.* two times the corresponding curvature parameter $\theta_i$.

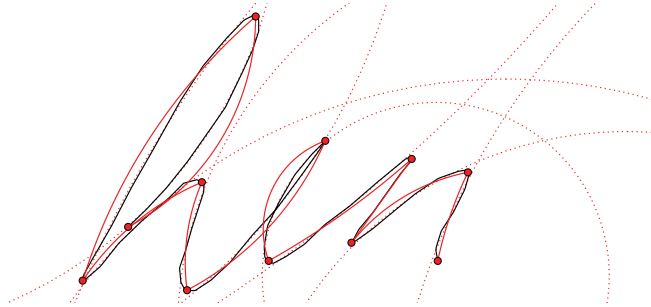

Figure 14: Fitting circles (dotted red) and circular arcs (red) to the input.

**Estimating stroke time-overlap parameters.** This step is based on the observation that a smaller values of $\Delta t_{0i}$, *i.e.* a greater time overlap between strokes, result in smoother trajectories. On the contrary, a sufficiently large value of $\Delta t_{0i}$ will result in a sharp corner in proximity of the corresponding virtual target. We exploit this notion, and compute an estimate of the $\Delta t_{0i}$ parameters by examining the sharpness of the input contour in the region of each key-point.

To do so we examine the previously computed turning angle surprisal signal, in which we can observe that sharp corners in the contour correspond with sharper peaks, while smoother corners correspond with smooth peaks with a larger spread. By treating the surprisal signal as a probability density function, we can then use statistical methods to measure the shape of each peak with a mixture of parametric distributions, and examine the shape of each mixture component in order to get an estimate of the corresponding sharpness along the input contour. To do so we employ a variant of Expectation Maximisation (EM) (Dempster et al., 1977) in which we treat the distance along the contour as a random variable weighted by the corresponding signal amplitude normalised to the $[0, 1]$ range. Once the EM algorithm has converged, we treat each mixture component as a radial basis function (RBF) centred at the corresponding mean, and use linear regression as in Radial Basis Function Networks (Stulp & Sigaud, 2015) to fit the mixture parameters to the original signal (Calinon, 2016). Finally we generate an estimate of sharpness $\lambda_i$ (bounded in the $[0, 1]$ range) for each key point using as a logarithmic function of the mixture parameters and weights. The corresponding $\Delta t_{0i}$ parameters are then given by

$$\Delta t_i = \Delta t_{min} + (\Delta t_{max} - \Delta t_{min})\lambda_i \ , \tag{9}$$

where $\Delta t_{min}$ and $\Delta t_{max}$ are user specified parameters that determine the range of the $\Delta t_{0i}$ estimates.

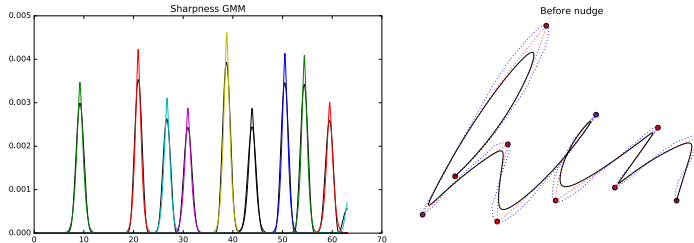

Figure 15: Sharpness estimation. Left, the GMM components estimated from the turning angle surprisal signal. Right, the $\Sigma\Lambda$ trajectory generated before the final iterative adjustment step. Note that at this stage the virtual target positions correspond with the estimated input key-points.

Note that we currently utilise an empirically defined function for this task. But in future steps, we intend to learn the mapping between sharpness and mixture component parameters from synthetically samples generated with the $\Sigma\Lambda$ model (for which $\Delta t_{0i}$, and consequently $\lambda_i$, are known).

**Iteratively estimating virtual target positions.** The loci along the input contour corresponding with the estimated key-points provide an initial estimate for a sequence of virtual targets, where each virtual target position is given by $\boldsymbol{v}_i = \boldsymbol{p}[\hat{z}_i]$. Due to the trajectory-smoothing effect produced by the time overlaps, the initial estimate will result in a generated trajectory that is likely to have a reduced scale with respect to the input we wish to reconstruct (Varga et al., 2005). In order to produce a more accurate reconstruction, we use an iterative method that shifts each virtual target towards a position that will minimise the error between the generated trajectory and the reconstructed input. To do so, we compute an estimate of $m$ *output* key-points $\{\boldsymbol{\xi}(z_i)\}$ in the generated trajectory, where $z_2, ..., z_m$ are the time occurrences at which the influence of one stroke exceeds the previous. These will correspond with salient points along the trajectory (extrema of curvature) and can be easily computed by finding the time occurrence at which two consecutive lognormals intersect. Similarly to the input key-point case, $\boldsymbol{\xi}(z_1)$ and $\boldsymbol{\xi}(z_m)$ respectively denote the first and last points of the generated trajectory. We then iteratively adjust the virtual target positions in order to move each generated

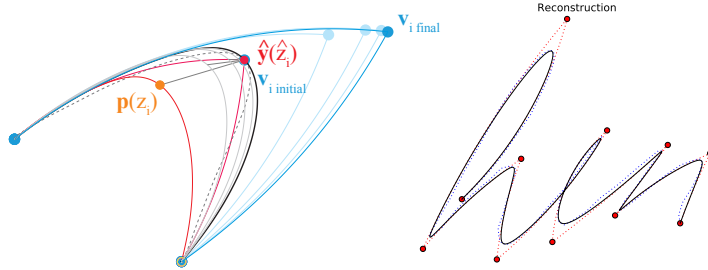

Figure 16: Final trajectory reconstruction step. Left, iterative adjustment of virtual target positions. Right, the final trajectory generated with the reconstructed dynamic parameters.

key-point $\boldsymbol{\xi}(z_i)$ towards the corresponding input key-point $\boldsymbol{p}[\hat{z}_i]$ with:

$$\boldsymbol{v}_i \leftarrow \boldsymbol{v}_i + \boldsymbol{p}[\hat{z}_i] - \boldsymbol{\xi}(z_i),\tag{10}$$

The iteration continues until the Mean Square Error (MSE) of the distances between every pair $\boldsymbol{p}[\hat{z}_i]$ and $\boldsymbol{\xi}(z_i)$ is less than an experimentally set threshold or until a maximum number of iterations is reached (Fig. 16). This method usually converges to a good reconstruction of the input within few iterations (usually $< 5$). Interestingly, even though the dynamic information of the input is discarded, the reconstructed velocity profile is often similar to the original (in number of peaks and shape), which can be explained by the extensively studied relationships between geometry and dynamics of movement trajectories (Viviani & Terzuolo, 1982; Lacquaniti et al., 1983; Viviani & Schneider, 1991; Flash & Handzel, 2007).

# C  RMDN MODEL DETAILS

In order to increase the expressive generative capabilities of our networks, we train them to model parametric probability distributions. Specifically, we use *Recurrent Mixture Density Networks* that output the parameters of a *bivariate Gaussian Mixture Model*.

## C.1  BIVARIATE RECURRENT MIXTURE DENSITY NETWORK

If a target variable $\boldsymbol{z}_t$ can be expressed as a *bivariate* GMM, then for $K$ Gaussians we can use a network architecture with output dimensions of $6K$. This output vector would then consist of $(\hat{\boldsymbol{\mu}}_t \in \mathbb{R}^{2K}, \hat{\boldsymbol{\sigma}}_t \in \mathbb{R}^{2K}, \hat{\boldsymbol{\rho}}_t \in \mathbb{R}^K, \hat{\boldsymbol{\pi}}_t \in \mathbb{R}^K)$, which we use to calculate the parameters of the

GMM via (Graves, 2013)

$$\boldsymbol{\mu}_t^k = \hat{\boldsymbol{\mu}}_t^k : \text{means for } k\text{'th Gaussian, } \boldsymbol{\mu}_t^k \in \mathbb{R}^2$$

$$\boldsymbol{\sigma}_t^k = \exp(\hat{\boldsymbol{\sigma}}_t^k) : \text{standard deviations for } k\text{'th Gaussian, } \boldsymbol{\sigma}_t^k \in \mathbb{R}^2$$

$$\rho_t^k = \tanh(\hat{\rho}_t^k) : \text{correlations for } k\text{'th Gaussian, } \rho_t^k \in (-1, 1) \tag{11}$$

$$\pi_t^k = \text{softmax}(\hat{\pi}_t^k) : \text{mixture weight for } k\text{'th Gaussian }, \sum_k^K \pi_t^k = 1$$

We can then formulate the probability distribution function $P_t$ at timestep $t$ as

$$P_t = \sum_k^K \pi_t^k N(\boldsymbol{z}_t \mid \boldsymbol{\mu}_t^k, \boldsymbol{\sigma}_t^k, \rho_t^k), \quad \text{where} \tag{12}$$

$$\mathcal{N}(\boldsymbol{x} \mid \boldsymbol{\mu}, \boldsymbol{\sigma}, \rho) = \frac{1}{2\pi\sigma_1\sigma_2\sqrt{1-\rho^2}} \exp\left[-\frac{Z}{2(1-\rho^2)}\right], \quad \text{and} \tag{13}$$

$$Z = \frac{(x_1 - \mu_1)^2}{\sigma_1^2} + \frac{(x_2 - \mu_2)^2}{\sigma_2^2} - \frac{2\rho(x_1 - \mu_2)(x_2 - \mu_2)}{\sigma_1\sigma_2} \tag{14}$$

## C.2  TRAINING OBJECTIVE

If we let $\boldsymbol{\theta}$ denote the parameters of a network, and given a training set $S$ of input-target pairs $(\boldsymbol{x} \in X, \hat{\boldsymbol{y}} \in \hat{Y})$, our training objective is to find the set of parameters $\boldsymbol{\theta}_{ML}$ which has the *maximum likelihood (ML)*. This is the $\boldsymbol{\theta}$ that maximises the probability of training set $S$ and is formulated as (Graves, 2008)

$$\boldsymbol{\theta}_{ML} = \arg\max_\theta \Pr(S \mid \boldsymbol{\theta}) \tag{15}$$

$$= \arg\max_\theta \prod_{(\boldsymbol{x}, \hat{\boldsymbol{y}})}^S \Pr(\hat{\boldsymbol{y}} \mid \boldsymbol{x}, \boldsymbol{\theta}). \tag{16}$$

Since the logarithm is a monotonic function, a common method for maximizing this likelihood is minimizing its negative logarithm, also known as the *Negative Log Likelihood (NLL)*, *Hamiltonian* or *surprisal* (Lin & Tegmark, 2016). We can then define our cost function $J$ as

$$J = -\ln \prod_{(\boldsymbol{x}, \hat{\boldsymbol{y}})}^S \Pr(\hat{\boldsymbol{y}} \mid \boldsymbol{x}, \boldsymbol{\theta}) \tag{17}$$

$$= -\sum_{(\boldsymbol{x}, \hat{\boldsymbol{y}})}^S \ln \Pr(\hat{\boldsymbol{y}} \mid \boldsymbol{x}, \boldsymbol{\theta}). \tag{18}$$

For a bivariate RMDN, the objective function can be formulated by substituting eqn. (12) in place of $\Pr(\hat{\boldsymbol{y}} \mid \boldsymbol{x}, \boldsymbol{\theta})$ in eqn. (18).

## C.3  V2V MODEL

**Input**  At each timestep $i$, the input to the V2V model is $\boldsymbol{x}_i \in \mathbb{R}^3$, where the first two elements are given by $\boldsymbol{\Delta v}_i$ (the relative position displacement for the $i$'th stroke, i.e. between the $i$'th virtual target and the next), and the last element is $u_i \in \{0, 1\}$ (the pen-up state during the same stroke). Given input $\boldsymbol{x}_i$ and its current internal state $(\boldsymbol{c}_i, \boldsymbol{h}_i)$, the network learns to predict $\boldsymbol{x}_{i+1}$, by learning the parameters for the Probability Density Function (PDF) : $\Pr(\boldsymbol{x}_{i+1} \mid \boldsymbol{x}_i, \boldsymbol{c}_i, \boldsymbol{h}_i)$. With a slight abuse of notation, this can be expressed more intuitively as $\Pr(\boldsymbol{x}_{i+1} \mid \boldsymbol{x}_i, \boldsymbol{x}_{i-1}, ..., \boldsymbol{x}_{i-n})$ where $n$ is the maximum sequence length.

**Output**  We express the predicted probability of $\boldsymbol{\Delta v}_i$ as a bivariate GMM as described in Section C.1, and $u_i$ as a Bernoulli distribution. Thus for $K$ Gaussians the network has output dimensions of $(6K + 1)$ which, in addition to eqn. (11), contains $\hat{e}_i$ which we use to calculate the pen state probability via (Graves, 2013)

$$e_i = \frac{1}{1 + \exp(\hat{e}_i)}, \quad e_i \in (0, 1) \tag{19}$$

**Architecture**  We use Long Short-Term Memory (Hochreiter & Schmidhuber, 1997) networks with input, output and forget gates (Gers et al., 2000), and we use Dropout regularization as described by Pham et al. (2014). We employ both a grid search and a random search (Bergstra & Bengio, 2012) on various hyperparameters in the ranges: sequence length {64, 128}, number of hidden recurrent layers {1, 2, 3}, dimensions per hidden layer {64, 128, 256, 400, 512, 900, 1024}, number of Gaussians {5, 10, 20}, dropout keep probability {50%, 70%, 80%, 90%, 95%} and peepholes {with, without}.

For comparison we also tried a deterministic architecture whereby instead of outputing a probability distribution, the network outputs a direct prediction for $\boldsymbol{x}_{i+1}$. As expected, the network was unable to learn this function, and all sequence of virtual targets synthesized with this method simply travel in a repeating zig-zag line.

**Training**  We use a form of Truncated Backpropagation Through Time (BPTT) (Sutskever, 2013) whereby we segment long sequences into *overlapping* segments of maximum length $n$. In this case long-term dependencies greater than length $n$ are lost, however with enough overlap the network can effectively learn a *sliding window* of length $n$ timesteps. We shuffle our training data and reset the internal state after each sequence. We empirically found an overlap factor of 50% to perform well, though further studies are needed to confirm the sensitivity of this figure.

We use *dynamic unrolling* of the RNN, whereby the number of timesteps to unroll to is not set at compile time, in the architecture of the network, but unrolled dynamically while training, allowing variable length sequences. We also experimented with *repeating* sequences which were shorter than the maximum sequence length $n$, to complete them to length $n$. We found that for our case they produced desirable results, with some side-effects which we discuss in later sections.

We split our dataset into training: 70%, validation: 20% and test:10% and use the Adam optimizer (Kingma & Ba, 2014) with the recommended hyperparameters. To prevent exploding gradients we clip gradients by their global L2 norm as described in (Pascanu et al., 2013). We tried thresholds of both 5 and 10, and found 5 to provide more stability.

We formulate the loss function $J$ to minimise the Negative Log Likelihood as described in Section C.2 using the probability density functions described in eqn. (12) and eqn. (19).

## C.4   V2D MODEL

**Input**  The input to this network at each timestep $i$ is identical to that of the V2V-model, $\boldsymbol{x}_i \in \mathbb{R}^3$, where the first two elements are $\boldsymbol{\Delta v}_i$ (normalised relative position displacement for the $i$'th stroke), and $u_i \in \{0, 1\}$ (the pen state during the same stroke). Given input $\boldsymbol{x}_i$ and its current internal state $(\boldsymbol{c}_i, \boldsymbol{h}_i)$, the network learns to predict the dynamic parameters $(\Delta t_{0i}, \theta_i)$ for the current stroke $i$, by learning the parameters for $\Pr(\Delta t_{0i}, \theta_i \mid \boldsymbol{x}_i, \boldsymbol{c}_i, \boldsymbol{h}_i)$. Again with an abuse of notation, this can be expressed more intuitively as $\Pr(\Delta t_{0i}, \theta_i \mid \boldsymbol{x}_i, \boldsymbol{x}_{i-1}, ..., \boldsymbol{x}_{i-n})$ where $n$ is the maximum sequence length.

**Output**  We express the predicted probability of the dynamic parameters $(\Delta t_{0i}, \theta_i)$ as a bivariate GMM as described in Section C.1.

**Architecture**  We explored very similar architecture and hyperparamereters as the V2V-model, but found that we achieved much better results with a shorter maximum sequence length. We trained a number of models with a variety of sequence lengths {3, ..., 8, 13, 16, 21, 32}.

**Training**  We use the same procedure for training as the V2V-model.

### C.5 A2D Model

**Input**  The input to this network $\boldsymbol{x}_i \in \mathbb{R}^5$ at each timestep $i$ is slightly different to the V2V and V2D models. Similar to the V2V and V2D models, the first two elements are $\boldsymbol{\Delta v}_i$ (normalised relative position displacement for the $i$'th stroke), and the third element is $u_i \in \{0, 1\}$ (the pen state during the same stroke). However in this case the final two elements are the dynamic parameters for the *previous stroke* $(\Delta t_{0i-1}, \theta_{i-1})$, normalized to zero mean and unit standard deviation.

Given input $\boldsymbol{x}_i$ and its current internal state $(\boldsymbol{c}_i, \boldsymbol{h}_i)$, the network learns to predict the dynamic parameters $(\Delta t_{0i}, \theta_i)$ for the current stroke $i$, by learning the parameters for $\Pr(\Delta t_{0i}, \theta_i \,|\, \boldsymbol{x}_i, \boldsymbol{c}_i, \boldsymbol{h}_i)$. Again with an abuse of notation, this can be expressed more intuitively as $\Pr(\Delta t_{0i}, \theta_i \,|\, \boldsymbol{x}_i, \boldsymbol{x}_{i-1}, ..., \boldsymbol{x}_{i-n})$ where $n$ is the maximum sequence length.

**Output**  The output of this network is identical to that of the V2D model.

**Architecture**  We explored very similar architecture and hyperparamereters as the V2D model.

**Training**  We use the same procedure for training as the V2V-model.

### C.6 Model selection

We evaluated and batch rendered the outputs of many different architectures and models at different training epochs, and settled on models which were amongst those with the lowest validation error, but also produced visibly more desirable results. Once we picked the models, the results displayed are not cherry picked.

The preprocessed IAM dataset contains 12087 samples (8460 in the training set) with maximum sequence length 305, minimum 6, median 103 and mean 103.9. For the V2V/V2D/A2V models trained on the IAM database we settle on an architecture of 3 recurrent layers, each with size 512, a maximum sequence length of 128, 20 Gaussians, dropout keep probability of 80% and no peepholes.

For the augmented one-shot learning models we used similar architectures, but found that 2 recurrent layers each with size 256 was able to generalise better and produce more interesting results that both captured the prime inputs without overfitting.

For V2V we used L2 normalisation on $\boldsymbol{\Delta v}_i$ input, and for A2D/V2D we used

We also tried a number of different methods for normalising and representing $\boldsymbol{\Delta v}_i$ on the input to the models. We first tried normalising the components individually to have zero mean and unit standard deviation. We also tried normalising uniformly on L2 norm again to have zero mean and unit standard deviation. Finally, we tried normalised polar coordinates, both absolute and relative.

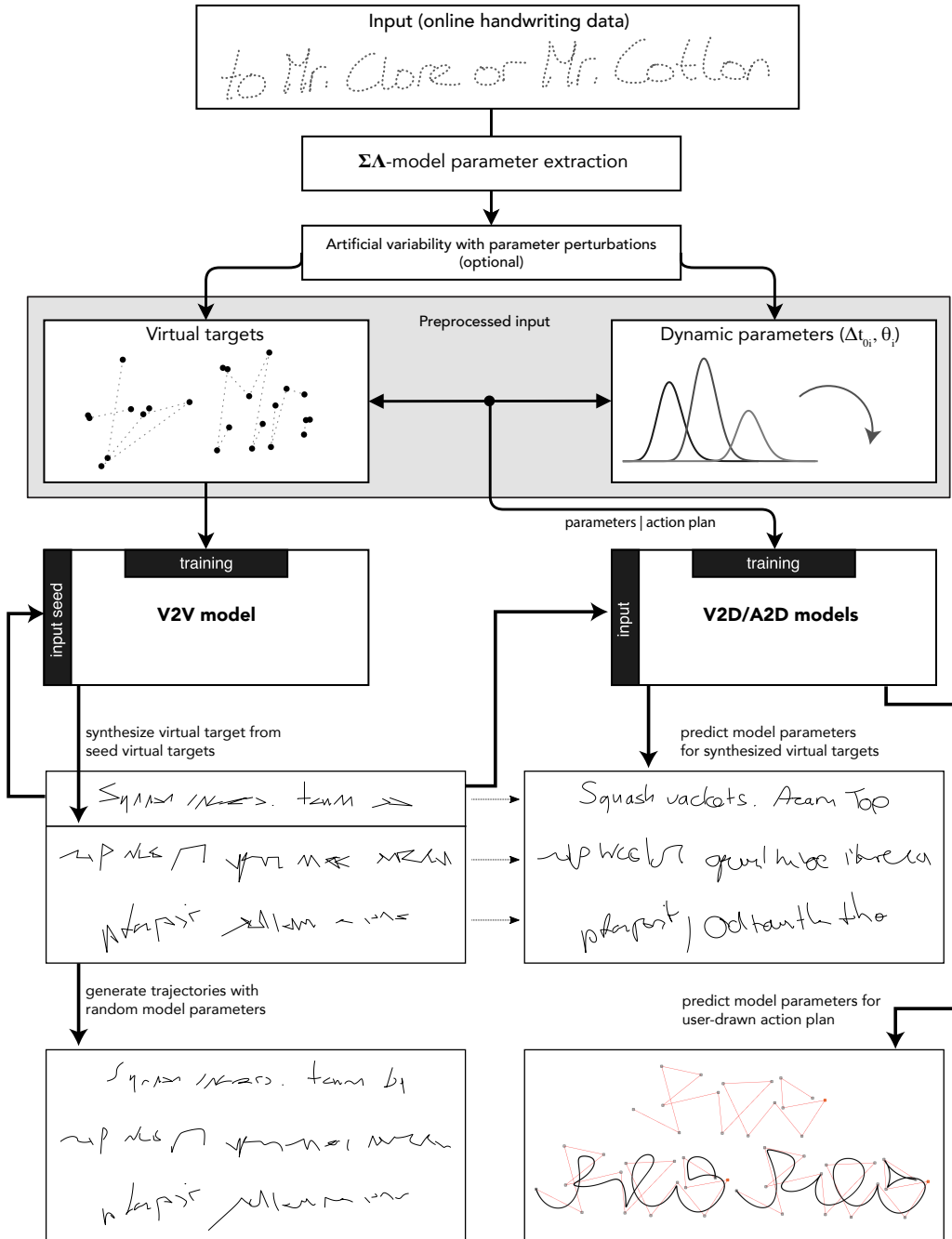

Figure 17: Schematic overview of the system.

