# Peer review of "Sequence generation with a physiologically plausible model of handwriting and Recurrent Mixture Density Networks"

_ICLR 2017 — rejected_

[Reviewer Comment · AnonReviewer2 · 07 Dec 2016]
**Offline/online**

The IAM dataset is an offline dataset (images). How did you generate the stroke sequences?

[Reviewer Comment · AnonReviewer2 · 07 Dec 2016]
**Sequence prediction**

Recent models exist for sequence prediction (from primed inputs) for various applications, e.g. for skeleton data. These models learn complex motion w/o any pre-processing. Did you look at the state of the art on this? Shouldn't this be applicable here?

[Official Review · AnonReviewer2 · rating 3 · confidence 5 · 16 Dec 2016]
**Interesting, but low novelty, and sub-standard evaluation**

The paper presents a method for sequence generation with a known method applied to feature extracted from another existing method. The paper is heavily oriented towards to chosen technologies and lacks in literature on sequence generation. In principle, rich literature on motion prediction for various applications could be relevant here. Recent models exist for sequence prediction (from primed inputs) for various applications, e.g. for skeleton data. These models learn complex motion w/o any pre-processing. 

Evaluation is a big concern. There is no quantitative evaluation. There is no comparision with other methods.

I still wonder whether the intermediate representation (developed by Plamondon et al.) is useful in this context of a fully trained sequence generation model and whether the model could pick up the necessary transformations itself. This should be evaluated.

Details:

There are several typos and word omissions, which can be found by carefully rereading the paper.

At the beginning of section 3, it is still unclear what the application is. Prediction of dynamic parameters? What for? Section 3 should give a better motivation of the work.

Concerning the following paragraph

"While such methods are superior for handwriting analysis and biometric purposes, we opt for a less precise method (Berio & Leymarie, 2015) that is less sensitive to sampling quality and is aimed at generating virtual target sequences that remain perceptually similar to the original trace. 
"
This method has not been explained. A paper should be self-contained.

The authors mentioned that the "V2V-model is conditioned on (...)"; but not enough details are given. 

Generally speaking, more efforts could be made to make the paper more self-contained.

[Official Review · AnonReviewer1 · rating 3 · confidence 3 · 16 Dec 2016]
**New representation space for a cut down version of an old model, with no quantitative results**

This paper takes a model based on that of Graves and retrofits it with a representation derived from the work of Plamondon. 
part of the goal of deep learning has been to avoid the use of hand-crafted features and have the network learn from raw feature representations, so this paper is somewhat against the grain. 

The paper relies on some qualitative examples as demonstration of the system, and doesn't seem to provide a strong motivation for there being any progress here. 
The paper does not provide true text-conditional handwriting synthesis as shown in Graves' original work. 

Be more consistent about your bibliography (e.g. variants of Plamondon's own name, use of "et al." in the bibliography etc.)

[Official Review · AnonReviewer3 · rating 3 · confidence 3 · 21 Dec 2016]
**Nice paper, but no machine learning contribution or evaluation.**

This paper has no machine learning algorithmic contribution: it just uses the the same combination  of LSTM and bivariate mixture density network as Graves, and the detailed explanation in the appendix even misses one key essential point: how are the Gaussian parameters obtained as a transformation of the output of the LSTM.
There are also no numerical evaluation suggesting that the algorithm is some form of improvement over the state-of-the-art.

So I do not think such a paper is appropriate for a conference like ICLR. The part describing the handwriting tasks and the data transformation is well written and interesting to read, it could be valuable work for a conference more focused on handwriting recognition, but I am no expert in the field.

[Final Decision · Program Chairs · 06 Feb 2017]
**ICLR committee final decision**

The paper presents a system, namely a recurrent model for handwriting generation. However it doesn't make a clear case for what contribution is being made, or convincing experimental comparisons. The reviews, while short, provide consistent suggestions and directions for how this work could be improved and reworked.